# Impact of the Immunomodulatory Factor Soluble B7-H4 in the Progress of Preeclampsia by Inhibiting Essential Functions of Extravillous Trophoblast Cells

**DOI:** 10.3390/cells13161372

**Published:** 2024-08-17

**Authors:** Yuyang Ma, Liyan Duan, Beatrix Reisch, Rainer Kimmig, Antonella Iannaccone, Alexandra Gellhaus

**Affiliations:** Department of Gynecology and Obstetrics, University of Duisburg-Essen, 45147 Essen, Germany; mayuyang0921@gmail.com (Y.M.); duanliyan1988@126.com (L.D.); beatrix.reisch@uk-essen.de (B.R.); rainer.kimmig@uk-essen.de (R.K.); antonella.iannaccone@uk-essen.de (A.I.)

**Keywords:** B7-H4, trophoblast, Akt, STAT3, preeclampsia, placenta

## Abstract

A key aspect of preeclampsia pathophysiology is the reduced invasiveness of trophoblasts and the impairment of spiral artery remodelling. Understanding the causes of altered trophoblast function is critical to understand the development of preeclampsia. B7-H4, a checkpoint molecule, controls a wide range of processes, including T-cell activation, cytokine release, and tumour progression. Our previous findings indicated that B7-H4 levels are elevated in both maternal blood and placental villous tissue during the early stages of preeclampsia. Here, we investigated the function of B7-H4 in trophoblast physiology. Recombinant B7-H4 protein was used to treat *human* SGHPL-5 extravillous trophoblast cells. Biological functions were investigated using MTT, wound healing, and transwell assays. Signalling pathways were analysed by immunoblotting and immunofluorescence. The functionality of B7-H4 was further confirmed by immunoblotting and immunohistochemical analysis in placental tissues from control and preeclamptic patients following therapeutic plasma exchange (TPE) or standard of care treatment. This study showed that B7-H4 inhibited the proliferation, migration, and invasion capacities of SGHPL-5 extravillous cells while promoting apoptosis by downregulating the PI3K/Akt/STAT3 signalling pathway. These results were consistently confirmed in placental tissues from preterm controls compared to early-onset preeclamptic placental tissues from patients treated with standard of care or TPE treatment. B7-H4 may play a role in the development of preeclampsia by inhibiting essential functions of extravillous trophoblast cells during placental development. One possible mechanism by which TPE improves pregnancy outcomes in preeclampsia is through the elimination of B7-H4 amongst other factors.

## 1. Introduction

Preeclampsia (PE), a hypertensive disorder affecting 2–8% of pregnancies, is characterised by impaired placentation and subsequent multi-organ dysfunction [1]. Despite the current lack of direct evidence, the most widely accepted mechanism for the pathogenesis of PE is the placental ischaemia hypothesis [2]. This hypothesis states that the pathogenesis of PE consists of two stages: the first stage is abnormal placentation in early pregnancy due to high-risk factors affecting extravillous trophoblast function, and the second stage is systemic endothelial and secondary organ dysfunction caused by an excess of anti-angiogenic factors [3]. Therefore, in PE, trophoblast function is impaired and vascularisation is restricted to the decidua or superficial myometrium [4,5]. This leads to placental hypoxia and stress, resulting in abnormal placental development and dysfunction, ultimately manifesting as PE.

Therapeutic plasma exchange (TPE) is an emerging treatment option for both severe early-onset PE and HELLP syndrome by removing harmful factors from the maternal circulation [6]. Our retrospective study showed that TPE prolonged the gestational age of 21 early-onset PE patients (23.75 ± 2.26 weeks) by a mean of 8.25 ± 5.97 days from the initiation of treatment, while standard of care treatment prolonged the gestational age of 20 early-onset PE patients (27.57 ± 2.68 weeks) by a mean of 3.14 ± 4.57 days [7]. TPE significantly decreased the circulating levels of the anti-angiogenic factors soluble fms-like tyrosine kinase-1 (sFlt-1) and soluble endoglin (sEng) in PE patients [7]. A subsequent study by our group also found a reduction in soluble B7-H4 (sB7-H4) levels after TPE treatment in patients with PE [8]. We therefore hypothesise that sB7-H4 plays a role in the progression of PE.

B7-H4, a B7 family immune checkpoint molecule, is a type I transmembrane protein [9] and was found to be highly expressed in placental villous tissue and lower in placental decidua in PE patients compared to normal controls, as shown in our previous publication [10]. Furthermore, serum levels of sB7-H4 were found to be elevated in women at increased risk of PE as well as in confirmed PE patients compared to normal pregnant women [11]. Our findings suggest that sB7-H4 could potentially serve as a predictive biomarker to identify women at high risk of PE. In this study, we aim to further investigate the impact of elevated sB7-H4 on trophoblast cell behaviour in PE to better understand the pathological mechanisms mediated by sB7-H4.

## 2. Materials and Methods

### 2.1. Cell Culture and Treatment

The SGHPL-5 trophoblast cell line was generously supplied by G. Whitley from the Department of Basic Medical Sciences at the University of St. George’s in London. SGHPL-5 cells were cultured in F-10 Ham medium (Invitrogen, Karlsruhe, Germany) complemented with 10% foetal bovine serum (Gibco, New York, NY, USA), 2 mM L-glutamine (Invitrogen, Karlsruhe, Germany), and 1% penicillin–streptomycin (Sigma-Aldrich, St. Louis, MO, USA), and maintained in the incubator at 37 °C with 5% CO_2_.

For the treatment with B7-H4 protein, SGHPL-5 cells were cultivated with recombinant *human* B7-H4 his-tagged protein (rhB7-H4; R&D systems, Minneapolis, MN, USA) at a concentration ranging of 0–0.5 μg/mL for varying times ranging from 0 to 72 h. The PI3K/Akt/STAT3 activator IL-6 (10 ng/mL) and PI3K/Akt/STAT3 inhibitor LY-294002 (50 μM) were used to treat cells for 30 min prior to rhB7-H4 exposure.

### 2.2. MTT Proliferation Assay

SGHPL-5 cells (6 × 10^3^ cells/well) were seeded in 96-well plates and then exposed to varying concentrations and durations of rhB7-H4. Cell viability was assessed using the MTT assay as described previously [12]. The inhibition rate was calculated using the formula: inhibition (%) = (1 − OD value of experimental group/OD value of control group) × 100%. Experiments were repeated three times.

### 2.3. Wound Healing and Transwell Assay

The wound healing assay was conducted for evaluating migration properties. SGHPL-5 cells were cultured in 6-well plates until 80–90% confluency. The cells were scratched with a 1 mL pipette tip. Wells were washed once with 1xDPBS. Fresh medium with rhB7-H4 (0, 0.1, 0.5 μg/mL) was added. Images were taken at 0 and 8 h using an Axiovert 25 microscope (ZEISS, Oberkochen, Germany). Image J2 (version 2.1.0; Rawak Software Inc., Stuttgart, Germany) was used to measure the scratched area. Wound healing percentage was calculated. Relative wound closure rate was calculated using the following formula: relative wound closure (%) = (initial wound area − wound area at 8 h)/initial wound area × 100%.

The transwell assay was used to analyse the invasion probability. A total of 5 × 10^4^ cells were seeded in serum-free medium in the upper compartment of a transwell chamber (Corning, Lowell, MA, USA). After incubation with different concentrations of rhB7-H4 protein for 24 h, the cells on the lower membrane were fixed with 4% paraformaldehyde and then stained with 0.1% crystal violet. Stained cells were counted in three fields per membrane using an Axiovert 25 microscope (20× objective) (ZEISS, Oberkochen, Germany). Data are reported as means  ±  SD from three independent experiments.

### 2.4. Immunofluorescence Staining

Cells on the coverslips were fixed with 4% paraformaldehyde for 20 min at room temperature, permeabilised with 0.1% Triton X-100 for 15 min, then blocked with 5% BSA for 20 min at 37 °C. Primary antibodies Ki67 (Abcam, Cambridge, UK; ab16667, 1:300) or cleaved caspase 3 (CST, Danvers, MA, USA; 9664s, 1:100) were added separately and incubated for 2 h at 37 °C, followed by staining with Alexa Fluor 488-conjugated donkey anti-*rabbit* antibody (Abcam, Cambridge, UK; ab150073, 1:200) for 1 h at 37 °C. Nuclei were counterstained with DAPI, and images were captured using confocal fluorescence microscopy (Leica SP5, Wetzlar, Germany) and photographed with LAS AF software version 4.0 (Leica, Wetzlar, Germany).

### 2.5. Patient Cohort

Pregnant women who delivered at the University Hospital Essen, Germany were recruited for placental tissue analysis. The study was approved by the Institutional Review Board of the University Hospital Essen (12-5212-BO), and all patients provided informed consent for the TPE treatment as well as for the placental study. The clinical characteristics of the patient cohort are listed in Appendix A. All pregnant women included in this study had singleton pregnancies, with gestational ages ranging from 22 + 4 weeks to 33 + 6 weeks. The study comprised 37 pregnant women: a preterm control group (*n* = 12) matched for gestational age, without PE or foetal growth restriction (FGR) (22 + 4–33 + 6 weeks), a PE group (*n* = 13) consisting of patients receiving standard-of-care treatment (23 + 5–30 + 0 weeks), and an early-onset PE group treated with TPE during pregnancy (PE + TPE) (*n* = 12) (23 + 2–28 + 5 weeks).

Complications unrelated to PE led to preterm birth in the control group due to the following reasons: imminent uterine rupture, placental abruption, placental abruption with CTG pathology, unexplained premature labour, glioblastoma, placenta previa with bleeding, amniotic infection with PROM, intrauterine foetal death, amniotic sac prolapse with premature labour, CTG pathology, and placenta increta. PE, defined by ISSHP guidelines [13], includes de novo hypertension and maternal organ dysfunction. Early-onset PE and PE + TPE groups had no other complications except PE, HELLP, and FGR. In the PE group, 6 patients were diagnosed with both PE and HELLP, while 7 patients were diagnosed with only PE. In the PE + TPE group, 4 patients were diagnosed with both PE and HELLP, while 8 patients were diagnosed with only PE.

### 2.6. Therapeutic Plasma Exchange

For the detailed TPE procedure, refer to our previous publication [7]. In brief, TPE was conducted at the Department of Nephrology, University Hospital Essen, Germany. A dual-lumen catheter was inserted into the internal jugular vein for TPE. The Spectra Optia centrifuge system (Terumo BCT, Inc., Lakewood, CO, USA) and COM TPE (Fresenius, Bad Homburg, Germany) were the primary devices used. Initially, plasma replacement involved fresh-frozen plasma (FFP). Protocol modifications, including the use of a 4% *human* albumin solution and partial FFP for patient plasma exchange, were implemented to reduce allergic reactions as treatment progressed.

### 2.7. Immunoblotting

Total protein from SGHPL-5 cells, placental villi, and decidua tissues were extracted using RIPA lysis buffer. Each sample containing 20 μg protein was separated by 4–15% protein gels (Bio-Rad, CA, USA) and then transferred to PVDF membranes (Bio-Rad, CA, USA). After blocking with 5% non-fat milk, the proteins were detected using the primary and secondary antibodies listed in Table 1.

### 2.8. Immunohistochemical Staining

The tissue sections underwent standard dewaxing procedures followed by antigen retrieval and blocking with 5% BSA. Subsequently, sections were immersed in 3% H_2_O_2_ in methanol for 10 min to neutralize endogenous peroxidase activity. Cleaved caspase 3 antibody (Cell Signaling Technology, 9664s; 1:100) was incubated overnight at 4 °C, followed by 1 h incubation with secondary antibody HRP-conjugated goat anti-*rabbit* IgG antibody (Invitrogen, G21234; 1:100) at 37 °C. To enhance the antibody signal, biotin and horseradish peroxidase-conjugated streptavidin were applied. Following this, sections were counterstained using hematoxylin, dehydrated using graded ethanol, cleared with xylene, mounted, and visualised under a Zeiss Axiophot microscope (Carl Zeiss, Oberkochen, Germany). Finally, digital images were captured using a Nikon DS-U1 digital camera equipped with NIS-Elements BR 3.0 software (Nikon, Tokyo, Japan). The mean optical density values (AOD) were obtained utilizing Image J2 (version 2.1.0; Rawak Software Inc., Stuttgart, Germany), along with the IHC Profiler plugin [14], averaging measurements from five fields of view for each placental sample.

### 2.9. Statistical Analysis

The placenta sample size was determined by G Power software (version 3.1.9.7; Franz Faul, Kiel, Germany). GraphPad Prism 8.0 (GraphPad Software Inc., San Diego, CA, USA) was utilised for data analysis. The Mann–Whitney test was applied for non-parametric independent two-group comparisons. Ordinary one-way ANOVA was employed for parametric multiple-group comparisons, while for non-parametric multiple-group comparisons, the Kruskal–Wallis test was used. Results are presented as means ± standard deviation. Significance levels are denoted as * *p* < 0.05, ** *p* < 0.01, and *** *p* < 0.001 for all statistical tests with a probability value (*p*-value) of 0.05 or less.

## 3. Results

### 3.1. B7-H4 Reduces the Proliferative, Migratory, and Invasion Capacity of SGHPL-5 Cells

To evaluate the effect of B7-H4 on trophoblast proliferation, cell viability was assessed using the MTT assay. SGHPL-5 cells were exposed to varying concentrations of rhB7-H4 (0, 0.05, 0.1, 0.2, 0.5 μg/mL) for 24, 48, and 72 h. Investigation of the proliferative capacity revealed a dose-dependent reduction in cell proliferation of SGHPL-5 cells induced by rhB7-H4 protein (Figure 1A). Notably, the highest inhibition rate was observed after 24 h of treatment with the highest concentration of 0.5 μg/mL rhB7-H4, resulting in an inhibition rate of 22.76% ± 7.25%. Conversely, inhibition rates decreased to 13.18% ± 4.42% and 8.35% ± 3.42% after 48 and 72 h, respectively. Statistical significance was achieved at concentrations of 0.1, 0.2, and 0.5 μg/mL rhB7-H4 after 24 h of treatment.

To assess the impact of B7-H4 on the migration and invasion abilities of SGHPL-5 cells, wound healing (Figure 1B) and transwell assays (Figure 1D) were conducted, respectively. SGHPL-5 cells were exposed to varying concentrations of rhB7-H4 protein (0, 0.1, 0.5 μg/mL), demonstrating a significant decrease in migration and invasion ability with increasing concentrations of rhB7-H4 (Figure 1C,E).

### 3.2. B7-H4 Reduces Cyclin D1, Induces p21 Protein Expression, and Promotes Apoptosis in SGHPL-5 Cells

Cyclin D1, a crucial cell cycle protein facilitating the G1-to-S phase transition, collaborates with Cyclin-dependent kinase (CDK) complexes to promote cell proliferation [15]. Conversely, p21 acts as a CDK inhibitor, impeding Cyclin–CDK complex activity and thereby regulating cell cycle progression [16]. These proteins, Cyclin D1 and p21, intricately modulate cell proliferation and are targets of multiple signalling pathways, including PI3K/Akt [17], and STAT3 [18]. Additionally, Cyclin D1 plays a pivotal role in enhancing cell invasive capacity [19]. Increased apoptosis of trophoblast cells, including extravillous trophoblasts (EVTs), is a notable feature in patients with PE [20]. Hence, cleaved caspase 3, an apoptosis activation marker, was utilised to measure apoptosis levels in trophoblast cells.

To unravel the impact of B7-H4 on trophoblast cell viability encompassing proliferation, migration, and apoptosis, SGHPL-5 cells were treated with various concentrations of rhB7-H4 (0, 0.1, 0.5 μg/mL) across different time intervals (0, 12, 24, 48 h). Immunoblot analysis was conducted to evaluate the expression levels of proliferation-associated proteins such as Cyclin D1 and p21, along with apoptosis-related protein cleaved caspase 3 (Figure 2A,B). With increasing rhB7-H4 concentrations and prolonged incubation times, a significant decrease in Cyclin D1 expression, associated with an elevation in p21 expression, was observed in SGHPL-5 cells compared to untreated cells (Figure 2C,D). Furthermore, cleaved caspase 3 expression exhibited an increasing trend upon rhB7-H4 protein treatment (Figure 2C,D).

Moreover, cellular immunofluorescence assays showed that treatment with 0.1 μg/mL rhB7-H4 for 24 h significantly downregulated the expression of the proliferation marker protein Ki67 (Figure 2E,F) while inducing expression of apoptosis-related protein cleaved caspase 3 (Figure 2E,F) in SGHPL-5 cells compared to untreated control. As depicted in Figure 2E, Ki67 was observed in the nucleus, while cleaved caspase 3 was predominantly localised in the cytoplasm of SGHPL-5 cells.

### 3.3. B7-H4 Downregulates the PI3K/Akt/STAT3 Signalling Pathway in SGHPL-5 Cells

B7-H4 modulates various cellular processes such as proliferation, migration, and apoptosis through multiple signalling pathways. Among these pathways, the PI3K/Akt and STAT3 signalling pathways have been extensively investigated in HeLa cells [21]. Moreover, STAT3 phosphorylation at the Tyr705 site is influenced by Mtor [22], a downstream target of the PI3K/Akt pathway. We have observed that B7-H4 significantly influenced the cell physiological behaviour (proliferation, migration, and apoptosis) of SGHPL-5 cells, with many of these alterations known to be regulated by the PI3K/Akt/STAT3 signalling cascade [23]. Consequently, we treated SGHPL-5 cells with different concentrations of rhB7-H4 protein and assessed changes in these signalling pathway-related proteins at different time points. The results showed that the ratios of p-PI3K (p110α) to PI3K, p-Akt (Ser473) to Akt, and p-STAT3 (Tyr705) to STAT3 were significantly decreased upon increasing rhB7-H4 concentration and time (Figure 3A–D).

### 3.4. IL-6, a PI3K/Akt/STAT3 Activator, Attenuated B7-H4-Induced Inhibitory Effects in SGHPL-5 Cells

Interleukin 6 (IL-6), a cytokine, activates the PI3K/Akt/STAT3 signalling pathway by binding to IL-6R/GP130 on the cell membrane [24]. In order to confirm the role of the PI3K/Akt/STAT3 signalling pathway in the inhibitory effects of rhB7-H4 on proliferation and migration, as well as triggering of apoptosis in SGHPL-5 cells, we investigated whether the activator of this signalling pathway, IL-6, could mitigate the inhibitory effects of rhB7-H4 on SGHPL-5 cells. Simultaneously, the PI3K inhibitor LY-294002 [25] was employed as a positive control to suppress the PI3K/Akt/STAT3 cascade signalling. Before being exposed to rhB7-H4 protein, SGHPL-5 cells underwent pre-incubation with IL-6 or LY-294002 for 30 min.

In comparison to SGHPL-5 cells treated solely with rhB7-H4 protein, pre-treatment of cells with IL-6 resulted in an increase of the phosphorylated forms: p-PI3K to PI3K, p-Akt to Akt, and p-STAT3 to STAT3 ratios, as seen in Figure 4A,C–E. Concurrently, the expression of the proliferation-related protein Cyclin D1 was upregulated, while p21 and the apoptosis-related protein cleaved caspase 3 were downregulated (Figure 4B,F–H).

### 3.5. Serum sB7-H4 Levels Vary among Control and PE Patients Receiving Standard of Care or TPE Treatment

In a study by us from Iannaccone et al. we revealed that TPE significantly reduced sFlt-1 and sEng serum levels in early-onset PE patients and significantly prolonged gestation in those patients (*n* = 20) by 8.25 ± 5.97 days [7]. Our preliminary results in those patients showed that TPE also has the potential to reduce serum levels of sB7-H4 [8]. Therefore, we analysed the effect of sB7-H4 on placenta/trophoblast cells by studying three groups of patients (PE treated with standard of care and PE treated with TPE as well as normal pregnant controls), which revealed different sB7-H4 serum levels in pregnant women. We conducted a retrospective study of a cohort containing 37 pregnant women, including 13 PE patients treated with standard of care (PE group), 12 PE patients treated with TPE (PE + TPE group), and 12 preterm controls (control group). The clinical characteristics are listed in Appendix A. Analysing the preliminary mean values of sB7-H4 in control, PE, and PE + TPE patients showed the following results: 5.362, 20.10, and 12.02 ng/mL, respectively.

### 3.6. B7-H4 May Suppress Trophoblast Cell Proliferation and Induce Apoptosis in PE Patients by Downregulating PI3K/Akt/STAT3 Signalling Pathway

It is well established that the placenta comprises two main components: the maternal part (decidua basalis) and the ffoetal part (chorionic villi). The maternal part is responsible for providing maternal blood and immune cells to support foetal growth and development. The ffoetal part, on the other hand, is connected to the maternal part through the chorionic tissue, obtains nutrients and oxygen from maternal blood, and eliminates metabolic wastes [26]. Therefore, we examined the levels of Cyclin D1 and p21 proteins in these two compartments of the placenta separately to gain a detailed insight into the effects of sB7-H4 on the foetal–maternal compartment of the placenta.

Immunoblotting results revealed that in placental chorionic villi (Figure 5A), the expression level of p21 of PE patients receiving standard of care was significantly higher compared to both the preterm control group (*p* = 0.0176) and the PE + TPE group (*p* = 0.0099) (Figure 5B). Conversely, Cyclin D1 did not reveal differences among the three groups (Figure 5C). Investigating decidua basalis tissue (Figure 5D), p21 displayed a similar trend: an increase in the PE group compared to the control group (*p* = 0.0061) and a decrease in the PE + TPE group compared to the PE group (*p* = 0.0021) (Figure 5E). The trend for Cyclin D1 was opposite to that of p21 (Figure 5C).

In addition, immunohistochemistry of placental sections demonstrated a significant attenuation in placental apoptosis-related protein cleaved caspase 3 in PE patients receiving standard-of-care treatment (*n* = 10), compared to both preterm controls (*n* = 12) and PE patients treated with TPE (*n* = 12) (Figure 5G,H).

To investigate whether changes occur in PI3K/Akt/STAT3 pathway-related proteins in the placentas of pregnant women treated with TPE or standard of care displaying different serum sB7-H4 levels, immunoblotting was performed to examine the levels of PI3K, Akt, STAT3, and the corresponding phosphorylated proteins in placental villi and decidual tissues of controls, PE patients treated with TPE, or standard of care treatment (Figure 6A,B).

In chorionic villi, our findings revealed a significant decrease in the ratios of p-PI3K/PI3K and p-Akt/Akt in PE patients compared to both the control group and the PE + TPE group (Figure 6C,D). However, the ratio of p-STAT3/STAT3 did not change among the three groups (Figure 6E). In the decidua basalis, while lacking statistical significance, the ratios of p-PI3K/PI3K and p-STAT3/STAT3 exhibited a decrease in the PE group compared to the control group, while an increase was observed in the PE + TPE group compared to the PE group (Figure 6F,H). There was no statistical difference in p-Akt/Akt changes among the three groups (Figure 6G).

These observations indicated distinct changes in the PI3K/Akt/STAT3 pathway in placental tissues across the control, PE, and PE + TPE groups, with notable differences in both chorionic villi and decidua basalis.

## 4. Discussion

The current study found that an increase in sB7-H4 suppresses extravillous SGHPL-5 trophoblast proliferation, migration, and invasion capacities in vitro, probably through downregulation of the PI3K/Akt/STAT3 pathway. In addition, ex vivo experiments further confirmed that sB7-H4 overexpression impairs placental development in PE placental tissues. In PE patients, TPE appears to improve placental function by indirectly activating the PI3K/Akt/STAT3 pathway through the elimination of sB7-H4. This provides new insights into the pathological mechanisms and potential therapeutic avenues for PE. The proper development of the placenta and its compartment lineages during the early stages is crucial for a successful pregnancy [27]. The disruption of cytotrophoblast (CTB) differentiation into EVTs hinders the invasion of trophoblasts into the uterus, resulting in inadequate remodelling of spiral arteries and reduced blood flow to the placenta [28,29]. Early-onset PE is particularly associated with inadequate early placental development, defined as occurring before 34 weeks’ gestation. During both the preclinical and clinical phases, maternal immune cells are also involved. Lymphocyte T cells have the ability to greatly enhance the production of inflammatory cytokines, such as tumour necrosis factor alpha (TNFα) and IL-6, as has been shown in PE [30,31]. Also, inflammation can induce and enhance trophoblast apoptosis. Apoptosis plays a vital role in safeguarding trophoblasts against maternal immune cell attacks and facilitating the replacement of uterine artery endothelial cells with EVTs in normal pregnancies [32,33]. However, during inflammation, the induction of apoptosis can have negative consequences for the trophoblast, causing major disruptions to trophoblast migration and placental vascularisation. This can also worsen immunological responses [34].

B7-H4, a member of the B7 family, serves as a checkpoint molecule that controls multiple functions including T-cell activation, cytokine release, tumour progression, and invasion capabilities [35,36,37]. We therefore hypothesise that B7-H4 plays a crucial role in the physiological processes of trophoblast cells. In the context of a healthy pregnancy, B7-H4 protein is widely expressed in CTB, syncytiotrophoblasts (STBs), EVTs, and mesenchymal stromal cells in the first trimester, with a particularly broad distribution in the STB cytoplasm and cell membrane, but its expression decreases during pregnancy [38]. We have confirmed that B7-H4 protein is still expressed in STBs and EVTs in late pregnancy [10]. However, in PE, B7-H4 changes are complex. Our previous research showed reduced B7-H4 expression in the decidua, while membrane-bound B7-H4 increased in the placental villi, leading to increased sB7-H4 levels in the maternal circulation [10,11]. In this study, we revealed that elevated sB7-H4 levels may inhibit the PI3K/Akt/STAT3 pathway in extravillous trophoblast cells, impairing proliferation, migration, and promoting apoptosis, which may ultimately contribute to the placental dysfunction observed in PE. Zhou et al. elucidated that B7-H4 knockdown activates the MAPK/ERK1/2 and JAK/STAT pathways and increases human embryonic stem cell-derived trophoblast invasion [39], which is consistent with our conclusion that B7-H4 inhibits SGHPL-5 trophoblast cells invasion.

B7-H4 is found to upregulate Cyclin D1 and Cyclin E in *human* embryonic kidney HEK293 cells, thereby promoting cell proliferation [40]. Dong et al. observed that B7-H4 knockdown in liver cancer cell lines (Huh-7 and HepG2) induces cell cycle arrest in the G0/G1 phase, resulting in decreased cell proliferation [41]. However, in Epstein–Barr virus-positive B-cell lymphoma cells, B7-H4 activation significantly reduces the growth, leading to cell cycle arrest in the G0/G1 phase. This effect is accompanied by the downregulation of CDK4/6 expression, upregulation of p21 expression at both protein and RNA levels, and suppression of CDK2 and Cyclin E/D expression, as well as the inhibition of the AKT pathway [42]. In this study, it was observed that B7-H4 attenuated trophoblast cell proliferation, concomitant with a decrease in Cyclin D1 expression and an increase in p21 expression. In conclusion, B7-H4 exhibits differential proliferative effects on different cell types, as shown in the above studies; however, its targeted proteins involved in proliferation share similarities. This could be attributed to the diverse nature of cell types. In addition, B7-H4 induced significant cell apoptosis in trophoblast cells, indicating that the inhibition of trophoblast growth by B7-H4 primarily depends on apoptosis induction.

Signalling pathways associated with B7-H4 include JAK/STAT3, PI3K/Akt, and CXCL12/CXCR4 [21]. The association between B7-H4 and the JAK/STAT3 pathway has been extensively documented in various diseases, including tumours and inflammatory conditions [43]. Reduced levels of p-STAT3/STAT3 have been reported not only in experimental preeclamptic rat placentas but also in *human* placental trophoblasts and decidual stromata from PE patients [44]. In addition, Christensen et al. observed a decrease in serum-induced endothelial STAT3 (Tyr 705) activation in PE, further highlighting its association with this disease [45]. Furthermore, Mo et al. reported that downregulation of the JAK/STAT3 pathway in the HTR-8/SVneo trophoblast cell line correlated with inhibition of cell proliferation and increased apoptosis [46]. In addition, Suman et al. found that STAT3 was closely associated with the invasive potential of JEG3 trophoblast cells [47]. In our study, although JAK was not examined, we found that B7-H4 decreased p-STAT3/STAT3 expression in trophoblast cells, resulting in decreased cell proliferation, migration, and promoted apoptosis in SGHPL-5 cells. Another upstream signalling pathway associated with STAT3 is PI3K/Akt, which we also investigated in this study. Xu et al. showed that annexin A4 promoted the invasion of HTR-8/SVneo and JEG3 trophoblast cells by activating the PI3K/Akt pathway [48]. Thus, activation of the PI3K/Akt signalling pathway appears to be positively correlated with cell proliferation and migration in trophoblast cells. In this study, we showed that rhB7-H4 decreased the protein levels of p-PI3K/PI3K and p-Akt/Akt in SGHPL-5 cells. The use of LY-294002, an inhibitor of PI3K, resulted in a decrease in the ratio of p-Akt/Akt and p-STAT3/STAT3, indicating that STAT3 functions as a downstream factor of PI3K in SGHPL-5 cells.

Several studies have shown that TPE is an effective treatment for improving the condition of early-onset PE [6,7,49] and HELLP syndrome [50]. Preliminary data showed that TPE can reduce serum sB7-H4 levels in PE patients, which may be one of the mechanisms by which TPE mediates prolongation of pregnancy. To further investigate the effect of B7-H4 on trophoblast cells in vivo, *human* placental tissues (preterm controls, PE patients receiving standard care or TPE treatment) were used. Studies on *human* placental tissues showed reduced proliferation and migration and increased apoptosis in patients with PE compared to placental tissues from normal pregnant women. The PE patients treated with TPE showed a trend towards improvement compared to the PE cases treated with standard care. These results suggest that sB7-H4 also appears to inhibit proliferation and invasion and promote apoptosis of *human* trophoblast cells in patients and may exacerbate the disease state observed in PE. We also examined the expression of proteins related to the PI3K/Akt/STAT3 signalling pathway in placental tissue from the control, PE, and PE + TPE groups. However, not all proteins involved in this signalling pathway showed significant differences. In particular, changes in p-STAT3/STAT3 levels in chorionic villi and p-Akt/Akt levels in decidua tissues were not significantly different between any of the patient groups analysed. This may be due to the small sample cohort or to the fact that in addition to sB7-H4, there may be other factors such as sFlt-1, sEng, and other inflammatory factors [51] that have an important impact on pathological changes in the placenta in PE.

Limitations of this study: Although the sample size was limited, our findings suggest to some extent the potential adverse effects of B7-H4 in PE, as well as the role of TPE in activating the PI3K/Akt/STAT3 pathway to improve placental function. Throughout the experiment, only the extravillous-immortalised trophoblast cell line SGHPL-5 was used. However, the *human* placenta experiment makes up for this shortcoming. Future research will focus on investigating the effects of B7-H4 on primary trophoblast cells and its immunomodulatory functions during pregnancy.

## 5. Conclusions

In conclusion, as shown in Figure 7, our current study highlights that B7-H4 attenuates the PI3K/Akt/STAT3 signalling pathway in SGHPL-5 cells and placental tissues from early-onset PE patients, leading to increased cell apoptosis, as evidenced by increased cleaved caspase 3 expression and inhibition of proliferation and migration capacities. Treatment with IL-6, a PI3K/Akt/STAT3 activator, partially alleviated the inhibitory effects of B7-H4 on trophoblast cells, accompanied by a decrease in cleaved caspase 3 expression. These findings suggest that targeting B7-H4 may be a promising approach for therapeutic intervention in the early progression of PE, affecting early trophoblast proliferation and invasion.

## Figures and Tables

**Figure 1 cells-13-01372-f001:**
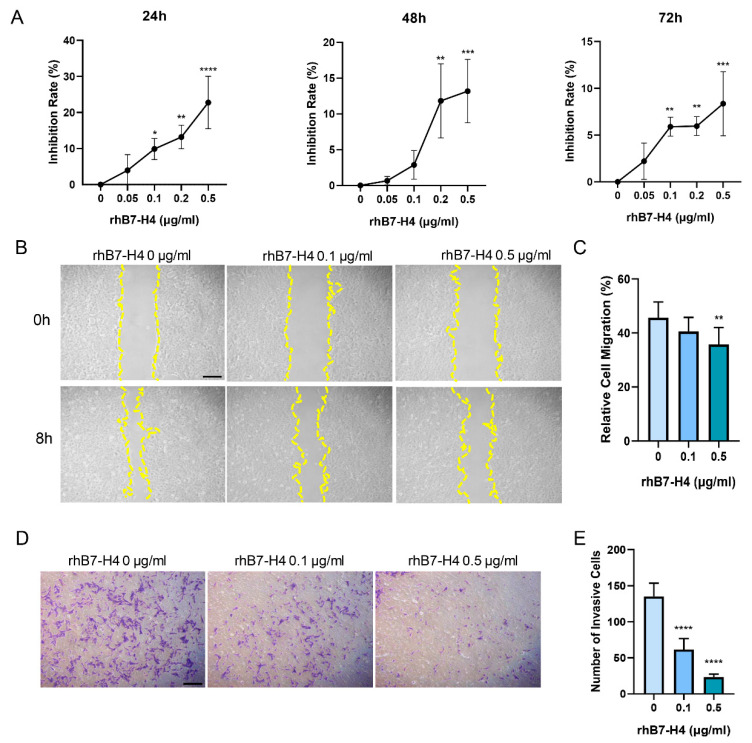
B7-H4 reduces the proliferative and migratory capacities of SGHPL-5 cells. (**A**) SGHPL-5 cells were treated with rhB7-H4 protein (0, 0.05, 0.1, 0.2 and 0.5 μg/mL) for 24, 48 or 72 h. Cell viability was determined by MTT assay. (**B**) The effect of B7-H4 on cell horizontal migration was determined by wound healing assay; yellow line: marked scratch. (**C**) Quantitative analysis of (**B**). (**D**) The effect of B7-H4 on cell invasion was determined by transwell assay. SGHPL-5 cells were treated with different concentrations of rhB7-H4 protein (0, 0.1, 0.5 μg/mL) for 24 h. (**E**) Quantitative analysis of (**D**). Data represent means ± SD of triplicate experiments. * *p*  <  0.05, ** *p*  <  0.01, *** *p*  <  0.001, **** *p* < 0.0001 vs. control. Scale bar: 1 mm.

**Figure 2 cells-13-01372-f002:**
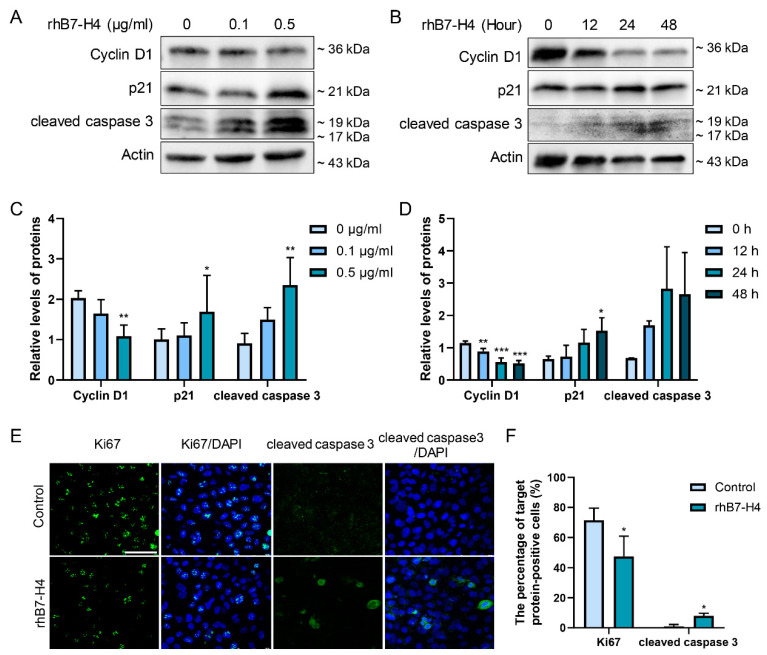
Effects of rhB7-H4 on the expression of Cyclin D1, p21, and cleaved caspase 3 proteins in SGHPL-5 cells. SGHPL-5 cells were treated with rhB7-H4 at various concentrations for 24 h (**A**) or with 0.1 μg/mL rhB7-H4 for 0, 12, 24, 48 h (**B**), and the levels of Cyclin D1, p21, cleaved caspase 3 and actin proteins were determined by immunoblotting. (**C**) Quantitative analysis of (**A**) to represent Cyclin D1/actin, p21/actin, and cleaved caspase 3/actin. (**D**) Quantitative analysis of (**B**) to represent Cyclin D1/Actin, p21/actin, and cleaved caspase 3/actin. (**E**) Effects of rhB7-H4 on the expression of Ki67 and cleaved caspase 3 in SGHPL-5 cells were determined by immunofluorescence (green: Ki67; blue: DAPI nucleus staining; merge: Ki67/DAPI). (**F**) Quantitative analysis of (**E**). Data represent means ± SD of triplicate experiments. Significance: * *p* < 0.05, ** *p* < 0.01, *** *p* < 0.001 versus control. Scale bar: 75 μm.

**Figure 3 cells-13-01372-f003:**
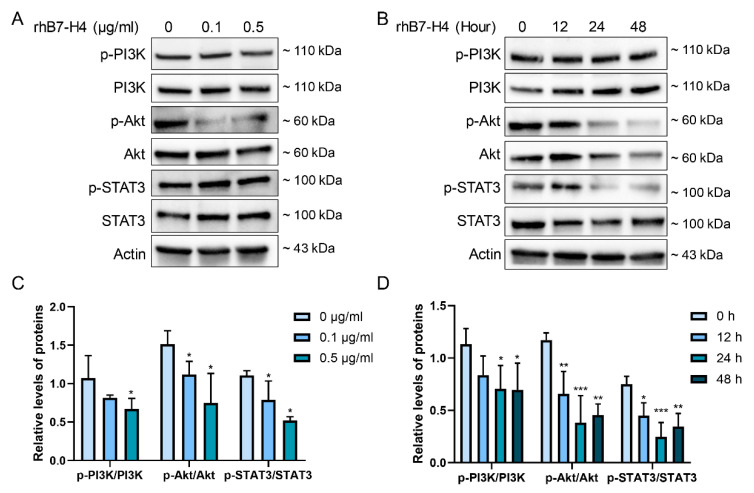
B7-H4 downregulates the PI3K/Akt/STAT3 signalling pathway in SGHPL-5 cells. (**A**) SGHPL-5 cells were incubated with rhB7-H4 at various concentrations for 24 h, then the PI3K/Akt/STAT3 signal pathway-related proteins and actin were determined by immunoblotting. (**B**) SGHPL-5 cells were incubated with 0.1 μg/mL rhB7-H4 for 0, 12, 24, 48 h, and the PI3K/Akt/STAT3 signal pathway-related proteins and actin were determined by immunoblotting. (**C**) Quantitative analysis of (**A**) to represent p-PI3K/PI3K, p-Akt/Akt, p-STAT3/STAT3. (**D**) Quantitative analysis of (**B**) to represent p-PI3K/PI3K, p-Akt/Akt, p-STAT3/STAT3. Data represent means ± SD of triplicate experiments. Significance: * *p* < 0.05, ** *p* < 0.01, *** *p* < 0.001 versus control.

**Figure 4 cells-13-01372-f004:**
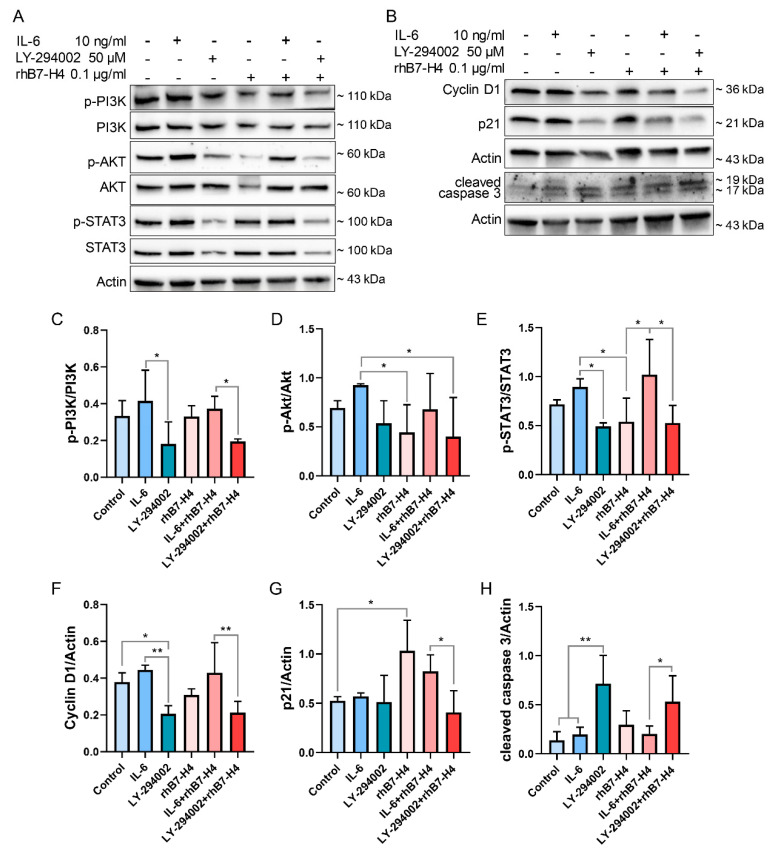
IL-6, an activator of PI3K/Akt/STAT3, partially attenuated B7-H4-induced inhibitory effects in SGHPL-5 cells. SGHPL-5 cells were incubated with rhB7-H4/IL-6/LY-294002 alone or in combination for 24 h, and the levels of PI3K/Akt/STAT3 pathway-related proteins, (**A**) as well as Cyclin D1, p21, and cleaved caspase 3 (**B**), were determined by immunoblotting. (**C**–**E**) Quantitative analysis of (**A**). (**F**–**H**) Quantitative analysis of (**B**). Data represent means ± SD of triplicate experiments. Significance: * *p* < 0.05, ** *p* < 0.01.

**Figure 5 cells-13-01372-f005:**
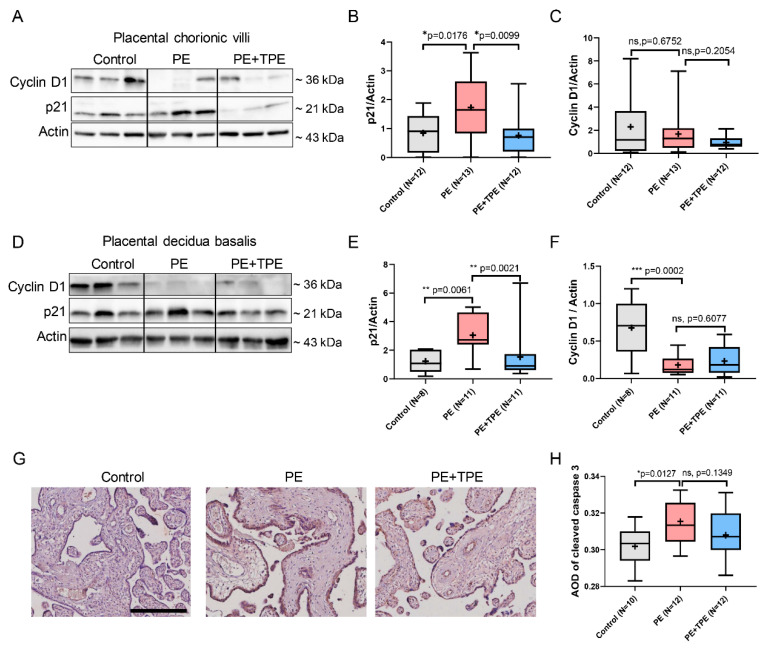
B7-H4 may suppress trophoblast cell proliferation and induce apoptosis in patients with PE. (**A**) Representative immunoblot of p21 and Cyclin D1 protein expression in placental chorionic villi of the control (*n* = 12), PE (*n* = 13) and PE + TPE (*n* = 12) groups. (**B**,**C**) Quantitative analysis of (**A**). (**D**) Representative immunoblot of p21 and Cyclin D1 protein expression in placental decidua basalis of the control (*n* = 8), PE *n* = 11) and PE +TPE (*n* = 11) groups. (**E**,**F**) Quantitative analysis of (**D**). (**G**) IHC staining images of cleaved caspase 3 in placental sections among control (*n* = 10), PE (*n* = 12), and PE + TPE (*n* = 12) patients. (**H**) Quantitative analysis of (**G**) to represent the AOD value of cleaved caspase 3 expression in placental tissue section. Data represent medians (lines inside boxes)/means (crosses inside boxes)  ±  interquartile ranges with minimum/maximum values as whiskers. Scale bar: 300 μm. Significance: * *p* < 0.05, ** *p* < 0.01, *** *p* < 0.001. ns: not significant.

**Figure 6 cells-13-01372-f006:**
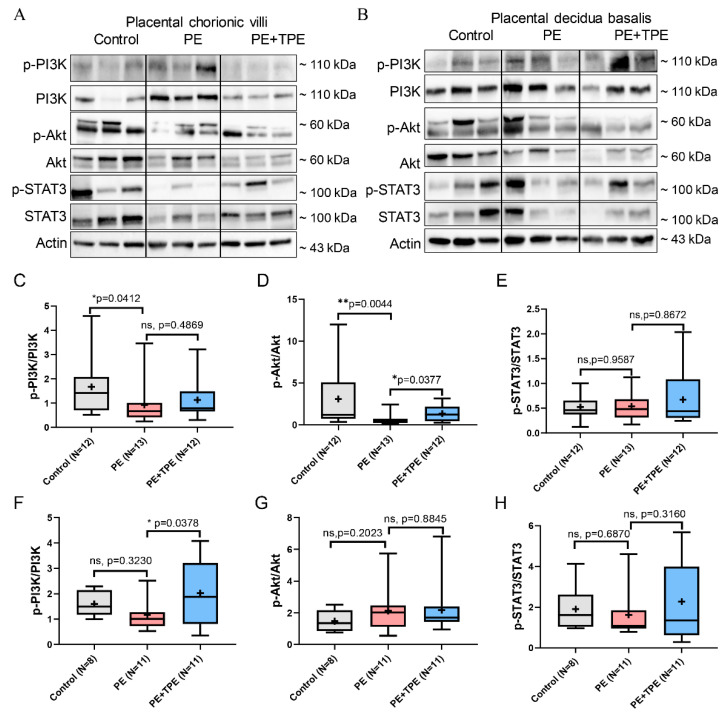
B7-H4 may suppress PI3K/Akt/STAT3 pathway in placental chorionic villi and decidua basalis in patients with PE. (**A**) Representative immunoblot of p-PI3K, PI3K, p-Akt, Akt, p-STAT3 and STAT3 protein expression levels in placental chorionic villi of the control (*n* = 12), PE (*n*= 13) and PE + TPE (*n* = 12) groups. (**B**) Representative immunoblot of p-PI3K, PI3K, p-Akt, Akt, p-STAT3 and STAT3 protein expression levels in placental decidua basalis of the control (*n* = 8), PE (*n* = 11) and PE + TPE (*n* = 11) groups. (**C**–**E**) Quantitative analysis of (**A**). (**F**–**H**) Quantitative analysis of (**B**). Data represent medians (lines inside boxes)/means (crosses inside boxes)  ±  interquartile ranges with minimum/maximum values as whiskers. Significance: * *p* < 0.05, ** *p* < 0.01. ns: not significant.

**Figure 7 cells-13-01372-f007:**
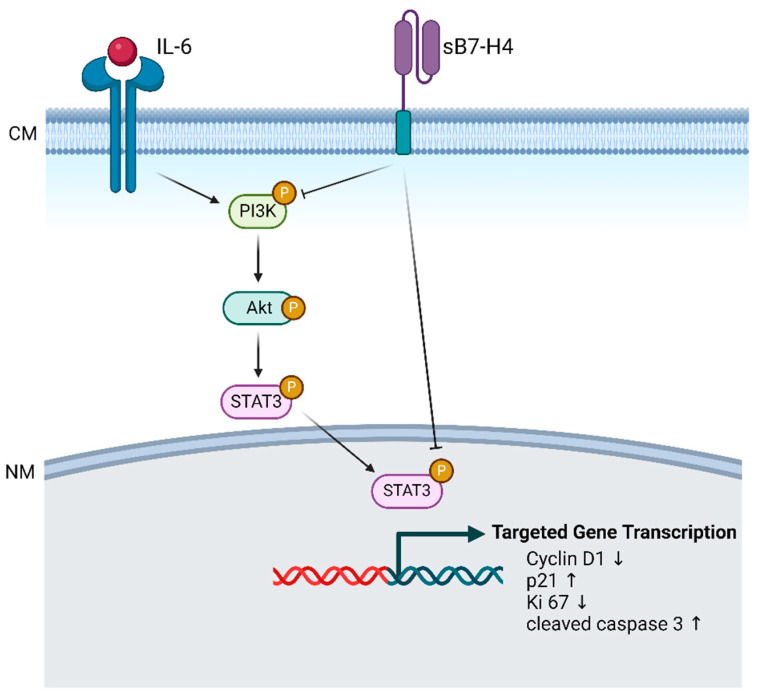
Potential schematic overview of the B7-H4-mediated signalling pathway in trophoblast cells in patients with early-onset PE. sB7-H4 diminishes trophoblast cell proliferation and migration while promoting apoptosis. The introduction of IL-6, an activator of the PI3K/Akt/STAT3 pathway, reverses sB7-H4’s inhibitory effects on trophoblast cells. These findings illuminate sB7-H4’s role in regulating trophoblast cell proliferation, migration, and apoptosis via the PI3K/Akt/STAT3 pathway in PE. **Abbreviations:** CM: cell membrane; NM, nuclear membrane. The diagram was created with BioRender.com.

**Table 1 cells-13-01372-t001:** Antibodies used for immunoblotting.

Antigen	Source	Supplier (Catalogue Number)	Dilution
**Primary antibody**			
Cyclin D1	*Rabbit*	Abcam (ab13417; Cambridge, UK)	1:150,000
p21	*Rabbit*	Cell Signaling Technology (2947s; Danvers, MA, USA)	1:1000
Cleaved caspase 3	*Rabbit*	Cell Signaling Technology (9664s; Danvers, MA, USA)	1:1000
p-PI3K (p110α)	*Rabbit*	Cell Signaling Technology (4249; Danvers, MA, USA)	1:1000
PI3K	*Rabbit*	Abcam (ab191606; Cambridge, UK)	1:1000
p-Akt (Ser473)	*Rabbit*	Cell Signaling Technology (4060s; Danvers, MA, USA)	1:1000
Akt	*Rabbit*	Cell Signaling Technology (9272s; Danvers, MA, USA)	1:1000
p-STAT3 (Tyr705)	*Rabbit*	Cell Signaling Technology (9145s; Danvers, MA, USA)	1:1000
STAT3	*Mouse*	BD Bioscience (610190; San Jose, CA, USA)	1:750
β-Actin	*Mouse*	Sigma-Aldrich (A3854; St. Louis, MO, USA)	1:100,000
**Secondary antibody**			
*Rabbit*, HRP	*Goat*	Invitrogen (G21234; Carlsbad, CA, USA)	1:5000
*Mouse*, HRP	*Goat*	Pierce (EJ66453; Rockford, IL, USA)	1:5000

## Data Availability

Upon reasonable request, the corresponding author can provide access to the datasets utilised and analysed in the current study.

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
