# Peer review of "Impact of the Immunomodulatory Factor Soluble B7-H4 in the Progress of Preeclampsia by Inhibiting Essential Functions of Extravillous Trophoblast Cells"

_cells, 2024, doi:10.3390/cells13161372_

Round 1

Reviewer 1 Report

Comments and Suggestions for Authors

This study investigated the effects of B7-B4 on trophoblast cell line SGHPL-5 in vitro and the differences in the signaling targets affected by B7-B4 in human placentas from early-onset preeclampsia treated with or without therapeutic plasma exchange (TPE) vs. healthy controls.

The study presented some interesting data towards the inhibitory effects of recombinant B7-B4 on trophoblast functions and variable effects of TPE on placental expressions/activation of PI3K/AKT/STAT3 and CDK2/p21.  However, the manuscript needs clarifications at least these listed below. 

First, "All patients given informed consent (12-5212-BO)" is insufficient for clear ethics for the clinical study especially TPE treatment. Please clarify whether the study was approved by IRB.

The in vitro studies are logically designed, data support the conclusions. However, the relationships between AKT (as a signaling kinase), p-STAT3 (transcription factor), and target genes (p21/CDK1) need to be clarified. For instance, does AKT activate STAT3 directly? Whether p21/CDK1 are STAT3 target genes? etc.

The measurements of PI3K/AKT/SATA3 and their phosphorylated forms, protein levels of p21, CDK1, and cleaved caspase3 in placental and decidual tissues from the three groups of patients are important data (Fig. 5).  However, these data do not support the authors' claim that B7-B4 regulates these targets in these patients in vivo directly as sera of the patients contain too many factors in addition to the increased soluble B7-B4 (supplemental table 1). To support this statement/conclusion, additional studies of placental/decidual explant cultures treated with/without rhB7-B4 will be needed at least.

Reviewer 2 Report

Comments and Suggestions for Authors

The manuscript entitled "Impact of the immunomodulatory factor soluble B7-H4 in the 2 progress of preeclampsia by inhibiting essential functions of 3 extravillous trophoblast cells' overall is interesting and gives new light on the role of ICP in controlling trophoblast invasion and survival as well as its role in PE. The authors determine signaling pathways associated with B7-H4: JAK/STAT3, PI3K/Akt. SGHPL-5 cells were cultured with  B7-H4 protein, and relevant methods were used to determine the proliferative, migratory, and invasion capacity of SGHPL-5 cells, as well as apoptosis. The authors measured serum sB7-H4 levels among control  (normal pregnant) and PE patients with standard care or TPE treatment. The authors conclude that B7-H4 may suppress trophoblast cell proliferation and induce apoptosis in PE patients by downregulating PI3K/Akt/STAT3 signaling pathway. 

The authors with relevant methods in well patients included cohorts confirmed their suspect that B7-H4 attenuates the PI3K/Akt/STAT3 signaling pathway in SGHPL-5 cells and placental tissues from early-onset PE patients, leading to increased cell apoptosis as evidenced by increased cleaved-caspase 3 expression and inhibition of proliferation and migration capacities.

They also point out that treatment with IL-6, a PI3K/Akt/STAT3 activator, partially alleviated the inhibitory effects of B7-H4 on trophoblast cells.

IT is important to publish the following reports promptly.

Author Response

We thank the reviewer for the recognition of our work and the positive evaluation of the study.

Round 2

Reviewer 1 Report

Comments and Suggestions for Authors

No more comments.